# Evaluating meta-analysis as a replication success measure

**Jasmine Muradchanian* \*, Rink Hoekstra, Henk Kiers, Don van Ravenzwaaij**

Behavioural and Social Sciences, University of Groningen, Groningen, the Netherlands

\* jasmine.muradchanian@gmail.com

## Abstract

### Background

The importance of replication in the social and behavioural sciences has been emphasized for decades. Various frequentist and Bayesian approaches have been proposed to qualify a replication study as successful or unsuccessful. One of them is meta-analysis. The focus of the present study is on the way meta-analysis functions as a replication success metric. To investigate this, original and replication studies that are part of two large-scale replication projects were used. For each original study, the probability of replication success was calculated using meta-analysis under different assumptions of the underlying population effect when replication results were unknown. The accuracy of the predicted overall replication success was evaluated once replication results became available using adjusted Brier scores.

### Results

Our results showed that meta-analysis performed poorly when used as a replication success metric. In many cases, quantifying replication success using meta-analysis resulted in the conclusion where the replication was deemed a success regardless of the results of the replication study.

### Discussion

We conclude that when using meta-analysis as a replication success metric, it has a relatively high probability of finding evidence in favour of a non-zero population effect even when it is zero. This behaviour largely results from the significance of the original study. Furthermore, we argue that there are fundamental reasons against using meta-analysis as a metric for replication success.

## Introduction

Trust in scientific knowledge is dependent on being able to replicate empirical results directly and independently [1]. In the social and behavioural sciences, the importance of replication

**Data Availability Statement:** The SCORE data and R code are not made available at the request of the SCORE team in order to prevent identification of the individual studies. For accessing the SCORE

data, the reader can contact the coordinators of the SCORE program through the following website: https://www.cos.io/score. The R code for the OSC [4] data supporting the conclusions of this article is included within the article. The OSC [4] dataset supporting the conclusions of this article is available in the OSF repository, https://osf.io/vdnrb [39]. We used the subset "final" from the "masterscript.R" (https://osf.io/vdnrb [39]).

**Funding:** The author(s) received no specific funding for this work.

**Competing interests:** The authors have declared that no competing interests exist.

was already emphasized decades ago [e.g., 2, p. 2], and recently, multiple replications in various scientific fields have been conducted [e.g., 1, 3–9].

Despite the alleged importance of replication studies, they seem to be relatively rare [e.g., 10]. The replication studies that have been conducted, however, did not show a strong alignment between the original studies and the replications. In fact, these replications have caused some to think that the social sciences are currently in a replication crisis or crisis of confidence [11]. The main evidence for this replication crisis comes from efforts such as the large replication project conducted by the Open Science Collaboration (OSC) in 2015, in which 100 original psychological studies were replicated. Similar replication projects have also been performed in other scientific disciplines such as economics [1], social sciences in general [6], experimental philosophy [7], and preclinical research in cancer biology [9]. Across these projects, some differences have been observed in fields with regard to replication success rate. Depending on the methods that were used in these projects to quantify replication success, the replication success rates were 36–68% in psychology [4], 61–78% in economics [1], 57–67% in social sciences in general [6], and 70–78% in experimental philosophy [7]. In preclinical research in cancer biology, for positive effects, 40% of replications were successful according to at least three out of five binary replication success metrics; for null effects, 80% of replications succeeded on this basis [9].

There are several different statistical methods and procedures available for determining whether replication results are in line with original results (Nosek and Errington [12] have discussed what replication is; an overview of different replication success metrics and their performance can be found, for example, in Muradchanian et al. [13]). One way to quantify replication success is to use the frequentist $p$-value [14, 15]. The Bayesian vessel for hypothesis testing, the Bayes factor (BF), has also been used to quantify replication success [e.g., 6]. Alternatively, two approaches that focus on estimation rather than hypothesis testing for quantifying replication success are "coverage" [4] and the prediction interval [16]. In addition, there are statistical methods that have been specifically designed in order to quantify replication success such as the replication Bayes factor [17], the Small Telescopes approach [18], and the sceptical $p$-value [19].

A more integrative way that has been used to approach the quantification of replication success is meta-analysis. Meta-analysis focusses on combining the original study and its replication attempt(s) in order to obtain an estimate of the pooled effect size and to quantify evidence for the effect of interest that takes into account all included studies simultaneously [e.g., 20]. Although meta-analysis has been used in various replication projects for quantifying replication success [e.g., 1, 4, 6, 9], the authors of these studies have expressed their concerns regarding publication bias and other biases that can affect the meta-analytic estimates [1, 4, 6, 9], and in turn replication success. Due to these biases, the effect sizes are likely to be inflated [e.g., 21], and replication success rate might well be overestimated [e.g., 13].

The large replication projects mentioned above have raised concerns about the reliability of the published scientific literature in the social and behavioural sciences. These results led to the question whether it is possible to predict overall replication success. One large-scale project that aimed to investigate exactly that, is the Collaborative Assessments for Trustworthy Science, or repliCATS, project, which is a structured expert elicitation method [22]. RepliCATS is part of the Systematizing Confidence in Open Research and Evidence, or SCORE, program [23]; the SCORE program creates and validates algorithms to give confidence scores for research claims [24]. Specifically, to study the viability of tools, teams collect claims from scientific papers in the social and behavioural sciences into a database; then, these teams create expert and machine produced estimates of credibility, followed by validating the estimates by creating evidence of reproducibility, robustness, and replicability [24]. Elicitation approaches

such as the repliCATS project (that result in precise evaluations of the replicability of research) can mitigate a part of burden on large replication projects, which are very resource intensive, and assist evaluating a larger proportion of the published literature. After these elicitation techniques are tested and calibrated, they could be included in peer review processes in order to improve evaluation before a paper gets published [23].

The repliCATS project is based on the IDEA protocol [22]. The IDEA protocol, which asks individuals to Investigate, Discuss, Estimate, and Aggregate, aims to improve the accuracy of individual judgements that are used to inform science [25]. The procedure goes as follows: for each claim from scientific papers to be assessed, experts are asked to investigate background information. Then, they are asked to give a first anonymous estimate of the probability regarding the degree of belief that the event will occur, and to give justifications for these estimates. Subsequently, they receive feedback about the differences between their individual and other experts' estimates, followed by discussing the differences in these opinions. Finally, they are expected to give a second anonymous estimate of the probability, but now taking into account the information that was obtained from the feedback round and the group discussion [25, 26].

The repliCATS project is an attempt to predict replication success using (aggregated) subjective beliefs. A different way to predict replication success could be through a statistical lens. For instance, if one were to quantify replication success through the results of a meta-analysis, it becomes possible to provide a-priori estimates of the chance of replication success under different models of the underlying population effect. Meta-analysis is a somewhat frequently used technique in replication research in order to decide whether a replication study is successful. For example, in OSC [4], the following was mentioned: "We evaluated reproducibility using significance and *P* values, effect sizes, subjective assessments of replication teams, and meta-analyses of effect sizes" (p. 2). Furthermore, in Camerer et al. [1], the following was mentioned: "There are different ways of assessing replication, with no universally agreed-upon standard of excellence [19–23]" (p. 1434). Then, they discussed the replication indicators (and their results) including meta-analysis: "The original and replication studies can also be combined in a meta-analytic estimate of the effect size [19]. As shown in Fig 1B, in the meta-analysis, 14 studies (77.8%) have a significant effect in the same direction as in the original study" (p. 1434). In the present study, we would like to get more insight into how meta-analysis functions in the context of replication research. To the best of our knowledge, this has not been studied yet. By taking the repliCATS project as inspiration, we would like to study this by making aggregate predictions using a statistical approach. Specifically, we would like to study how meta-analysis functions as a replication success metric by making aggregate predictions regarding replicability under different assumptions about the population when one includes the original study and its single replication in the meta-analysis.

We used real original studies that would be truly replicated within the SCORE program. For each original SCORE study, we calculated the probability that the replication study would lead to a successful replication under different assumptions about the population. Note that replication results were not yet available so as not to influence the models we proposed for prediction. The probability of replication success can be perceived as a kind of statistical power (a meta-analysis replication power), conditional on the results of the original study. A successful replication in the present context means that the meta-analytic result is statistically significant (i.e., $p < .05$), and that the meta-analytic effect size is in the same direction as the original effect size. Accuracy of the predicted overall replication success was evaluated by comparing the observed and predicted percentage of replication successes. Because the number of studies we could use from the SCORE project was small, we decided to examine how well the models that we generated to predict replication success in the SCORE program would have predicted replication success for study results that were part of the OSC [4] project.

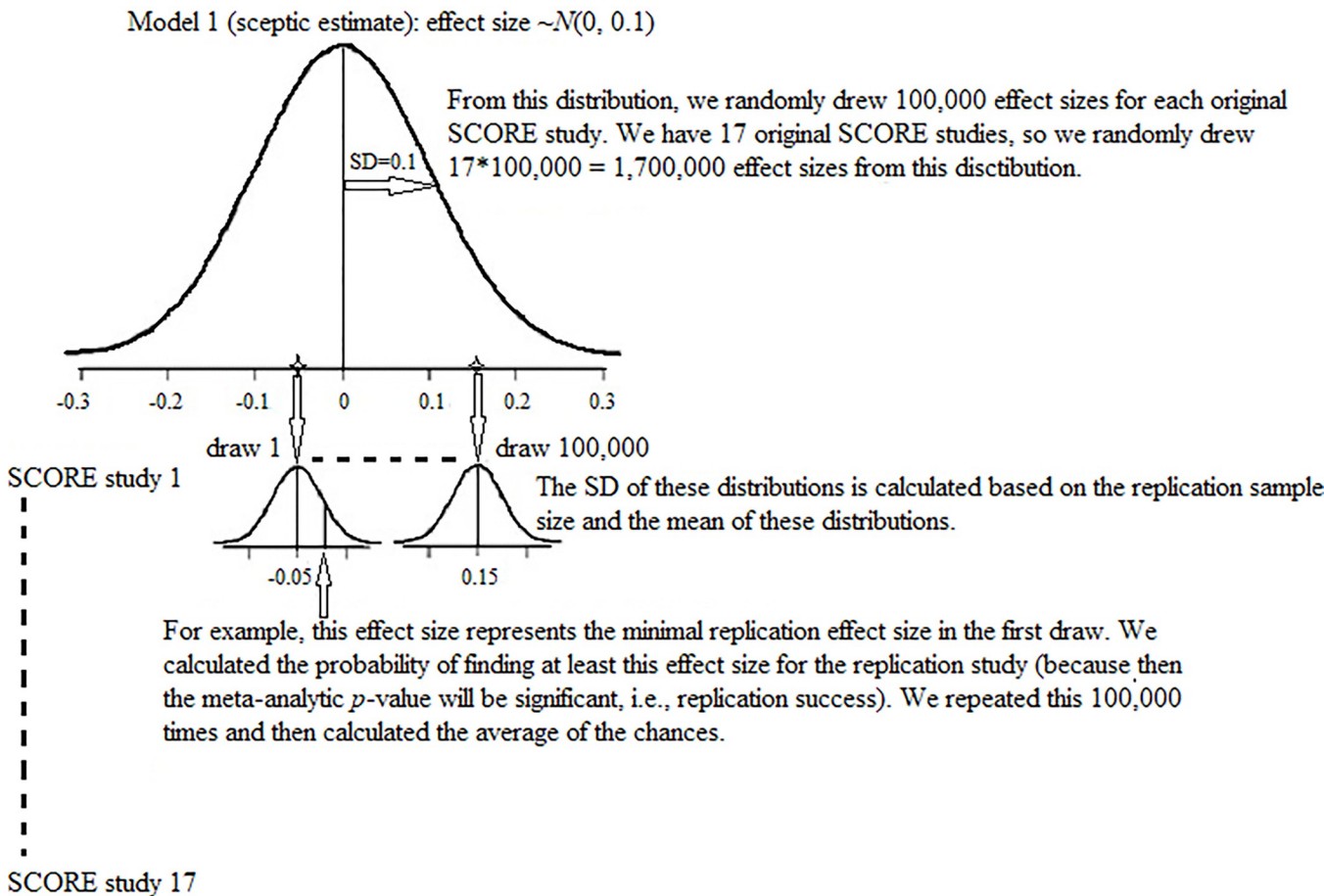

**Fig 1. Determination of the probability of finding the minimal replication effect size for Model 1.**

The structure of the rest of this paper is as follows: In the Method section, we describe step-by-step the set-up of this study. In the Results section, we look at the probability that original study results will be successfully replicated under different assumptions about the population. In the Discussion section, we discuss implications for replication researchers, including a discussion of the usefulness of meta-analysis as measure for replication success.

## Method

### Original studies

Results from real original studies that have been replicated within the SCORE program were used. From the SCORE authors, we received a data set with results from 81 original studies in total, published between 2009 and 2018, representing different journals in social and behavioural sciences. To perform our analyses, we needed the sample size, the observed effect size, and the type of the observed effect size from the original studies.

For each of these variables, the data set included an entry for values reported in the paper and/or values calculated by the SCORE team. An entry for 'values reported in the paper' could have been taken directly from the text of the original article or have been derived from the text. Values calculated by the SCORE team have been determined by the SCORE statistical consultants who used all available sources to present the most accurate representation of the key

quantity of interest. Our default was to use the values calculated by the SCORE team. For cases where these values were not available, we used the values reported in the paper. This was done in order to have as many complete values as possible for the variables of interest.

In the SCORE data set, some of the effect sizes (original studies) were positive and some were negative. Since the sign of the outcome was irrelevant for the purpose of our study, we decided to take the absolute values of all effect sizes of the original studies. Furthermore, the effect sizes of the original studies in the SCORE data set had various outcome measures such as correlation, partial correlation, Cramer's V, Phi, Cohen's $f^2$, partial eta$^2$, Cohen's d for independent samples, and Cohen's d for one sample or paired samples. The rest of the original SCORE studies had an unknown outcome measure. We restricted our analyses to studies with the following outcome measures: partial correlation, correlation, Cohen's d for independent samples, and Cohen's d for one sample or paired samples (n = 17). The disciplines of these 17 studies were: Business & Economics; Environmental Sciences & Ecology; Psychology; Behavioral Sciences; Biomedical Social Sciences; Sociology. We decided not to include studies with Cohen's $f^2$ (n = 17) and partial eta$^2$ (n = 1) as outcome measure because such measures are not ideal for meta-analysis, as they are directionless [W. Viechtbauer, personal communication, July 26, 2022]. Furthermore, we decided not to include studies with Cramer's V (n = 2) and Phi (n = 1) as outcome measure because such measures seem to be used rarely in practice.

The OSC [4] data set consisted of 73 original effects in total, published in 2008, representing three journals in psychology. We used the subset where all effect sizes were Fisher z transformed positive (partial) correlations.

## General procedure

After receiving the original SCORE studies in May 2022, the first step in our analysis was to determine how large the replication effect size should minimally be for a successful replication, the minimal replication effect size. A successful replication in our case refers to a set of studies for which the meta-analytic result is statistically significant ($\alpha$ = .05, one-sided), and for which the meta-analytic effect size is in the same direction as the original effect size [13]. In other words, our first step was to determine how large the replication effect size should minimally be for a statistically significant meta-analytic result. The meta-analyses were based on an original study and a single replication study, and we assumed that the true population effect size for the original and replication study was the same. Therefore, all meta-analyses were fixed-effect. Furthermore, we assumed that the sample size of the replication study was twice as large as the sample size of the original study, since replication studies tend to have a larger sample size than the original attempts [e.g., 1, 4, 6, 7]. To determine how large the replication effect size should minimally be for a significant meta-analysis, we conducted a meta-analysis on the known original effect size and a grid of replication effect sizes varying from -0.99 to 0.99 in increments of 0.001 (i.e., 1981 replication effect sizes) and retained the minimum effect size out of the grid that led to a significant result.

The meta-analyses were performed using the function *rma* in the R package *metafor* [27]. In our case, we needed to specify the following four elements in the *rma* function: effect size original study, effect size replication study, variance original study, and variance replication study. We specify the formulas and simplifications of parameters used in the *rma* function per outcome measure in Table 1. Since the distribution of (partial) correlation is not Normal, we applied Fisher z transformations. The combined meta-analytic effect size in the *rma* function was computed as follows in the context of our study (i.e., one original study and a single

**Table 1. Formulas and simplified parameters used in the rma function for each outcome measure.**

| Outcome measure | Fisher z transformation | Variance outcome measure | Simplified parameters in rma function |
|---|---|---|---|
| Correlation | $z_r = 0.5*\ln((1+r)/(1-r))$ | $Vz_r = \frac{1}{N-3}$ | $yi = z_{ro}$, replication effect sizes |
| | | | $vi = Vz_{ro}, Vz_{rr}$ |
| Partial correlation | $z_{pr} = 0.5*\ln((1+pr)/(1-pr))$ | $Vz_{pr} = \frac{1}{N-3-C}$, C is the number of control variables [29] | $yi = z_{pro}$, replication effect sizes |
| | | | $vi = Vz_{pro}, Vz_{prr}$ |
| Cohen's d: independent samples | N/A | $Vd_2 = \frac{n1+n2}{n1*n2} + \frac{d^2}{2*(n1+n2)}$ [30] | $yi$ = original effect sizes, replication effect sizes |
| | | | $vi = Vd_{2o}, Vd_{2r}$ |
| Cohen's d: one sample or paired samples | N/A | $Vd_1 = \left(\frac{1}{N} + \frac{d^2}{2*N}\right)*2*(1-r)$, r is the correlation between pre-measurement and post-measurement [30] | $yi$ = original effect sizes, replication effect sizes |
| | | | $vi = Vd_{1o}, Vd_{1r}$ |

*Note*: $z_{(p)r}$ represents Fisher z transformed (partial) correlation; ln represents the natural logarithm; (p)r represents the (partial) correlation; N represents the sample size in case of one sample; n1 and n2 represent the sample size in case of 2 samples (independent samples); yi represents the outcome measure of the original and replication study in the *rma* function; vi represents the variance of the original and replication study in the *rma* function; subscript $_o$ refers to the original study; subscript $_r$ refers to the replication study; for the SCORE studies that had partial correlation as outcome measure, we did not know the number of control variables, so we decided to use C = 0.

replication study):

$$M_{overall} = \frac{\frac{1}{Vo}*Mo + \frac{1}{Vr}*Mr}{\frac{1}{Vo} + \frac{1}{Vr}},$$ (1)

where $M_{overall}$ is the combined meta-analytic effect size, $V_o$ is the variance of the original study, $M_o$ is the observed effect size in the original study, $V_r$ is the variance of the replication study, and $M_r$ is the observed effect size in the replication study. The standard error of $M_{overall}$ can be computed as [28]

$$SE_{Moverall} = \sqrt{\frac{1}{\frac{1}{Vo} + \frac{1}{Vr}}}.$$ (2)

After determining how large the replication effect size should minimally be for a successful replication (see above), the next step was to determine what the probability is of finding the minimal replication effect size (for a significant meta-analytic result), given a specific model for the population effect sizes. In the present study, we used the following 3 models, which are specified in Table 2: the sceptic estimate (Model 1: population effect sizes are centered around zero), the moderate estimate (Model 2: population effect sizes are centered around half the sample effect size), and the optimistic estimate (Model 3: population effect sizes are centered around the sample effect size).

**Table 2. Three models included in the prediction of replication success.**

| Model | Model specification |
|---|---|
| 1) Sceptic estimate | effect size $\sim N(0, 0.1)$ |
| 2) Moderate estimate | effect size $\sim N(0.5*$effect size original study$, 0.1)$ |
| 3) Optimistic estimate | effect size $\sim N($effect size original study$, 0.1)$ |

For each of the 17 SCORE studies, we randomly drew 100,000 population effect sizes from each of the 3 models (distributions). For each of these, we assumed the sample effect size to be normally distributed, with the mean equal to the randomly drawn population effect size, and the variance as given by the formulas for the variances of effect sizes, for the corresponding outcome measures as provided in Table 1, where N was taken equal to two times the sample size of the original study. For each drawn population effect size, we now calculated the probability to find a replication effect size surpassing the value minimally required to obtain a significant meta-analysis outcome, given the generated sampling distribution of the replication sample effect size. Then, for each of the three models, we computed the probability of replication success for each SCORE study as the average probability of finding the minimal replication effect size over the 100,000 distributions per model. This resulting probability can be viewed as the statistical power to successfully replicate an original finding, in terms of a meta-analysis based measure. Here we will use it as a prediction phrased as the probability of successful meta-analysis based replication. In Fig 1, we visually illustrate this procedure for Model 1.

After the authors of the SCORE program completed the replication studies, they shared the results with us in January 2023, and we compared our predictions with their observed outcomes. The SCORE team defined replication success as a significant effect in the replication study ($\alpha$ = .05, two-tailed), combined with the replication effect size being in the same direction as the original effect size. Additionally, we also used meta-analysis to quantify replication success for 15 SCORE studies (two studies were excluded from this computation because the replication studies had a different outcome measure than the original studies). In the meta-analysis, we included the original SCORE study and its observed replication study. Replication success was established if the meta-analytic result was significant (i.e., $p < .05$), and before transformation, the meta-analytic effect size was in the same direction as the original effect size.

The accuracy of the (probabilistically) predicted overall replication success was evaluated by comparing the true and predicted percentage of replication successes using Brier scores [31, 32]. For each model separately, the Brier score was calculated as follows:

$$Brier\ score = \frac{\sum_{i=1}^{N}(predicted - observed)^2}{N} \qquad (3)$$

The predicted scores represent the probability that the original study will lead to a successful replication, given one of the three models. The observed scores represent the observed replication success; this can be either 0 (= the original study did not replicate successfully) or 1 (= the original study did replicate successfully). The N represents the number of studies. We computed Brier scores based on observed replication success such as assessed by the SCORE team where the number of studies was 17. Additionally, we computed Brier scores based on observed replication success assessed through meta-analysis where the number of studies was 15.

To interpret the Brier score, one should realize that the smaller the Brier score, the more accurate the prediction. Furthermore, a prediction probability of .5 represents pure chance prediction, and the associated Brier score of .25, represents chance performance. We followed Starns et al. [33] for enhancing interpretability by adjusting the Brier score such that –1 represents the worst possible performance, 0 represents chance performance, and 1 represents perfect performance. This means that scores at chance performance (i.e., Brier score = 0.25) became 0. Scores better than chance performance (i.e., Brier score < 0.25) got a value between 0 and 1. Scores worse than chance performance (i.e., Brier score > 0.25) got a value between -1

and 0. Our formula for the adjusted Brier score is as follows:

$$adjusted\ Brier\ score = 1 - 2*\sqrt{Brier\ score}, \tag{4}$$

where *Brier score* represents the original Brier score such as described previously.

We applied the same procedure to the OSC [4] data. Quantification of replication success for the OSC [4] replication studies was kept the same as the quantification of replication success for the SCORE replication studies.

## Results

The results for the original SCORE studies are presented in Table 3, and the results for the original OSC [4] studies are presented in Fig 2. The exact results for the OSC [4] original studies can be found in Table 4. The first column in Table 3 represents a study identifier. The second column in Table 3 represents the type of outcome measure the 17 original studies had in the SCORE data set, the third column represents the approximate observed effect size of these studies, and the fourth column provides the approximate sample size of these studies. For the OSC [4] data, the corresponding information can be found in columns 1, 2, and 3 in Table 4. The type of effect size is not included for the OSC [4] data set in Table 4, because all outcome measures were Fisher z transformed correlations, except for one outcome measure, which was a Fisher z transformed partial correlation.

The fifth column in Table 3 represents the minimal replication effect size, meaning that the replication study of the corresponding original SCORE study should have at least that effect

**Table 3. Information and results for the original SCORE studies.**

| Study identifier | Type of outcome measure | Effect size original SCORE study | | Sample size original SCORE study | | Minimal effect size replication study | p(Model 1: Sceptic estimate) | p(Model 2: Moderate estimate) | p(Model 3: Optimistic estimate) |
|---|---|---|---|---|---|---|---|---|---|
| | | ILB | IUB | ILB | IUB | | | | |
| SCORE_study01 | partial correlation | 0.2 | 0.3 | 250 | 300 | -0.02 | $0.58_3$ | $0.88_2$ | $0.98_1$ |
| SCORE_study02 | correlation | 0.2 | 0.3 | 60 | 80 | 0.04 | $0.38_4$ | $0.76_2$ | $0.96_1$ |
| SCORE_study03 | correlation | 0.4 | 0.5 | 100 | 150 | -0.11 | $0.81_2$ | $1.00_1$ | $1.00_1$ |
| SCORE_study04 | correlation | 0.4 | 0.5 | 80 | 100 | -0.1 | $0.78_2$ | $1.00_1$ | $1.00_1$ |
| SCORE_study05 | partial correlation | 0.5 | 0.6 | 100 | 150 | -0.16 | $0.91_1$ | $1.00_1$ | $1.00_1$ |
| SCORE_study06 | Cohen's d (indep) | 0.2 | 0.3 | 2200 | 2300 | -0.08 | $0.78_2$ | $0.98_1$ | $1.00_1$ |
| SCORE_study07 | Cohen's d (indep) | 0.4 | 0.5 | 1600 | 1700 | -0.15 | $0.92_1$ | $1.00_1$ | $1.00_1$ |
| SCORE_study08 | Cohen's d (indep) | 0.4 | 0.5 | 100 | 150 | 0.05 | $0.39_4$ | $0.87_2$ | $0.99_1$ |
| SCORE_study09 | Cohen's d (indep) | 0.5 | 0.6 | 100 | 150 | 0.02 | $0.46_4$ | $0.93_1$ | $1.00_1$ |
| SCORE_study10 | Cohen's d (indep) | 0.5 | 0.6 | 1800 | 1900 | -0.21 | $0.98_1$ | $1.00_1$ | $1.00_1$ |
| SCORE_study11 | Cohen's d (indep) | 0.6 | 0.7 | 100 | 150 | -0.04 | $0.60_3$ | $0.99_1$ | $1.00_1$ |
| SCORE_study12 | Cohen's d (indep) | 0.8 | 0.9 | 30 | 50 | 0.1 | $0.35_4$ | $0.88_2$ | $0.99_1$ |
| SCORE_study13 | Cohen's d (indep) | 1.5 | 1.6 | 20 | 40 | -0.06 | $0.58_3$ | $1.00_1$ | $1.00_1$ |
| SCORE_study14 | Cohen's d (indep) | 2.1 | 2.2 | 20 | 40 | -0.19 | $0.75_2$ | $1.00_1$ | $1.00_1$ |
| SCORE_study15 | Cohen's d (indep) | 2.9 | 3 | 500 | 550 | -0.62 | $1.00_1$ | $1.00_1$ | $1.00_1$ |
| SCORE_study16 | Cohen's d (one) | 0.4 | 0.5 | 100 | 150 | -0.05 | $0.65_3$ | $0.98_1$ | $1.00_1$ |
| SCORE_study17 | Cohen's d (one) | 0.6 | 0.7 | 40 | 60 | -0.06 | $0.65_3$ | $0.99_1$ | $1.00_1$ |

*Note*: Power above .9, power between .7 and .9, power between .5 and .7, and power under .5 are highlighted using subscript. Within the three types of outcome measures, the studies are sorted on the effect size of the original SCORE study. Note that exact values of effect sizes and sample sizes have been replaced with intervals at the request of the SCORE team in order to prevent identification of the individual studies. 'ILB' and 'IUB' refer to 'interval lower bound' and 'interval upper bound'.

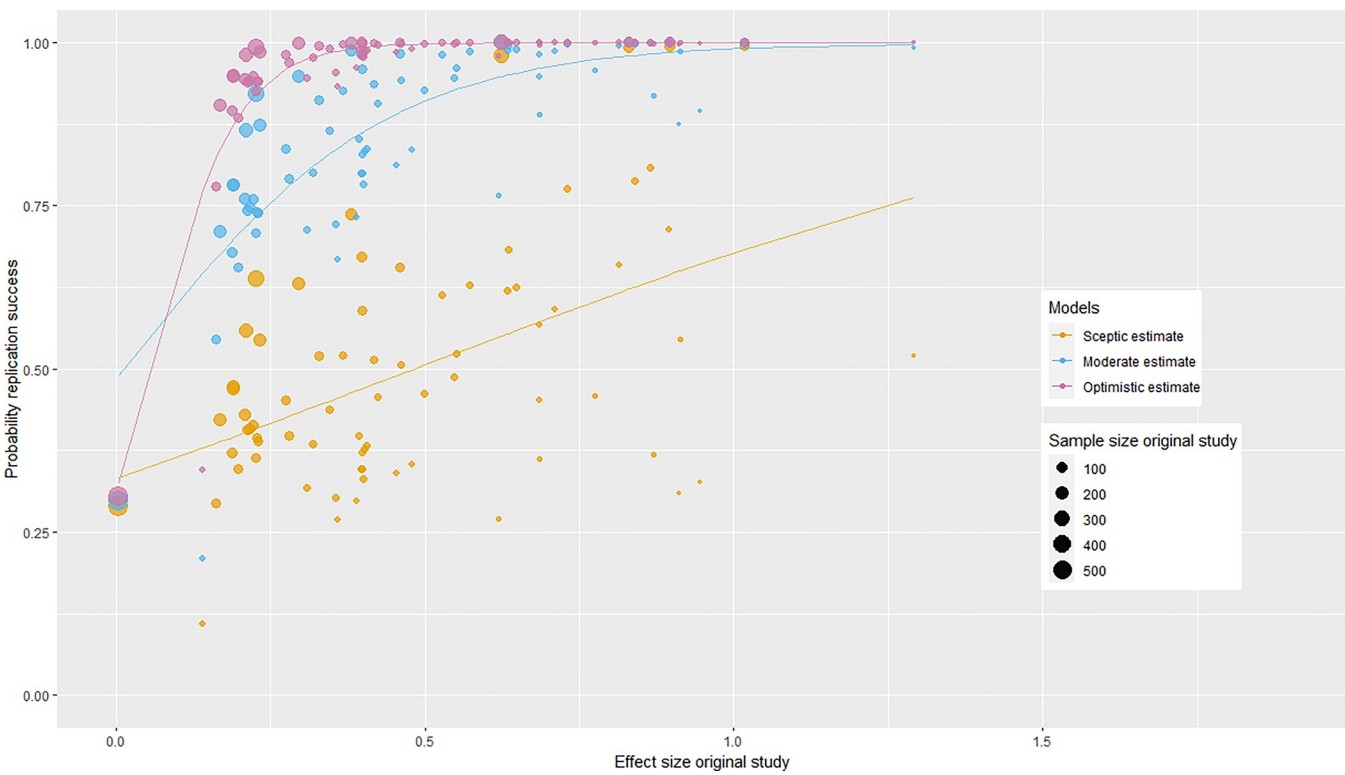

**Fig 2. Results for the original OSC [4] studies.**

size in order for the meta-analytic result to be statistically significant (i.e., the original study is then replicated successfully). In this column, it can be observed that most minimal replication effect sizes are negative (13 out of 17 replication effect sizes). This means that from a meta-analytic perspective, 13 out of 17 studies would be counted as a replication success even if a sample effect size of zero or lower were obtained. For the OSC [4] data, this information can be found in column 4, Table 4. Here, it can be observed that 31 out of 73 minimal replication effect sizes are negative, meaning that from a meta-analytic perspective, slightly fewer than half of these would be counted as a replication success even with a replication effect size of zero.

The probability of finding at least the minimal replication effect size for the 17 original SCORE studies, given Model 1 (i.e., the sceptic estimate model, where the true effect size $\sim N(0, 0.1)$) is presented in column 6 of Table 3. We can observe that these probabilities are pretty high for most original SCORE studies: 13 out of 17 probabilities are higher than 50%, meaning that according to our analysis, 13 out of 17 original SCORE studies have a probability of more than 50% to be successfully replicated, given the sceptic estimate model. The other four studies have probabilities higher than 30%. Taken together, this shows that if each of these studies have an underlying population effect of zero, it is still likely for many of these replications to be successful when viewed through a meta-analytic lens.

Regarding the OSC [4] data, 31 out of 73 probabilities are higher than 50% under the sceptic model (i.e., 31 of 73 orange dots are above 0.50), meaning that when using meta-analysis as a replication success metric, approximately half of the original OSC [4] studies (31 out of 73) had a probability of more than 50% to be successfully replicated, given the sceptic estimate model (see also column 5 in Table 4). The results here are somewhat less extreme than those for the SCORE studies.

**Table 4. Results for the OSC [4] studies.**

| Study identifier | Effect size original OSC (4) study | Sample size original OSC(4) study | Minimal effect size replication study | p(Model 1) | p(Model 2) | p(Model 3) |
|---|---|---|---|---|---|---|
| OSC_study01 | 0 | 564 | 0.06 | 0.29 | 0.3 | 0.3 |
| OSC_study02 | 0.14 | 28 | 0.21 | 0.11 | 0.21 | 0.35 |
| OSC_study03 | 0.16 | 96 | 0.07 | 0.29 | 0.55 | 0.78 |
| OSC_study04 | 0.17 | 184 | 0.02 | 0.42 | 0.71 | 0.9 |
| OSC_study05 | 0.19 | 118 | 0.04 | 0.37 | 0.68 | 0.9 |
| OSC_study06 | 0.19 | 196 | 0.01 | 0.47 | 0.78 | 0.95 |
| OSC_study07 | 0.19 | 196 | 0.01 | 0.47 | 0.78 | 0.95 |
| OSC_study08 | 0.2 | 96 | 0.05 | 0.35 | 0.66 | 0.88 |
| OSC_study09 | 0.21 | 133 | 0.02 | 0.43 | 0.76 | 0.94 |
| OSC_study10 | 0.21 | 259 | -0.02 | 0.56 | 0.87 | 0.98 |
| OSC_study11 | 0.21 | 117 | 0.03 | 0.41 | 0.74 | 0.94 |
| OSC_study12 | 0.22 | 113 | 0.03 | 0.41 | 0.75 | 0.94 |
| OSC_study13 | 0.22 | 110 | 0.03 | 0.41 | 0.76 | 0.95 |
| OSC_study14 | 0.23 | 375 | -0.04 | 0.64 | 0.92 | 0.99 |
| OSC_study15 | 0.23 | 86 | 0.04 | 0.36 | 0.71 | 0.93 |
| OSC_study16 | 0.23 | 96 | 0.03 | 0.39 | 0.74 | 0.94 |
| OSC_study17 | 0.23 | 94 | 0.03 | 0.39 | 0.74 | 0.94 |
| OSC_study18 | 0.23 | 192 | -0.01 | 0.54 | 0.87 | 0.99 |
| OSC_study19 | 0.28 | 92 | 0.02 | 0.45 | 0.84 | 0.98 |
| OSC_study20 | 0.28 | 70 | 0.03 | 0.4 | 0.79 | 0.97 |
| OSC_study21 | 0.3 | 174 | -0.04 | 0.63 | 0.95 | 1 |
| OSC_study22 | *0.31 | 43 | 0.07 | 0.32 | 0.71 | 0.95 |
| OSC_study23 | 0.32 | 53 | 0.04 | 0.38 | 0.8 | 0.98 |
| OSC_study24 | 0.33 | 85 | -0.01 | 0.52 | 0.91 | 1 |
| OSC_study25 | 0.35 | 57 | 0.02 | 0.44 | 0.86 | 0.99 |
| OSC_study26 | 0.36 | 33 | 0.08 | 0.3 | 0.72 | 0.96 |
| OSC_study27 | 0.36 | 28 | 0.11 | 0.27 | 0.67 | 0.93 |
| OSC_study28 | 0.37 | 69 | -0.01 | 0.52 | 0.93 | 1 |
| OSC_study29 | 0.38 | 154 | -0.07 | 0.74 | 0.99 | 1 |
| OSC_study30 | 0.39 | 28 | 0.09 | 0.3 | 0.73 | 0.96 |
| OSC_study31 | 0.39 | 40 | 0.04 | 0.4 | 0.85 | 0.99 |
| OSC_study32 | 0.4 | 33 | 0.06 | 0.35 | 0.8 | 0.98 |
| OSC_study33 | 0.4 | 33 | 0.06 | 0.35 | 0.8 | 0.98 |
| OSC_study34 | 0.4 | 75 | -0.03 | 0.59 | 0.96 | 1 |
| OSC_study35 | 0.4 | 36 | 0.05 | 0.37 | 0.83 | 0.99 |
| OSC_study36 | 0.4 | 101 | -0.05 | 0.67 | 0.98 | 1 |
| OSC_study37 | 0.4 | 31 | 0.07 | 0.33 | 0.78 | 0.98 |
| OSC_study38 | 0.4 | 36 | 0.05 | 0.38 | 0.83 | 0.99 |
| OSC_study39 | 0.41 | 36 | 0.05 | 0.38 | 0.84 | 0.99 |
| OSC_study40 | 0.42 | 53 | -0.01 | 0.51 | 0.94 | 1 |
| OSC_study41 | 0.42 | 43 | 0.02 | 0.46 | 0.91 | 1 |
| OSC_study42 | 0.45 | 26 | 0.07 | 0.34 | 0.81 | 0.99 |
| OSC_study43 | 0.46 | 70 | -0.05 | 0.66 | 0.98 | 1 |
| OSC_study44 | 0.46 | 43 | 0 | 0.51 | 0.94 | 1 |
| OSC_study45 | 0.48 | 25 | 0.07 | 0.35 | 0.84 | 0.99 |
| OSC_study46 | 0.5 | 33 | 0.02 | 0.46 | 0.93 | 1 |

*(Continued)*

**Table 4.** (Continued)

| Study identifier | Effect size original OSC (4) study | Sample size original OSC(4) study | Minimal effect size replication study | p(Model 1) | p(Model 2) | p(Model 3) |
|---|---|---|---|---|---|---|
| OSC_study47 | 0.53 | 46 | -0.04 | 0.61 | 0.98 | 1 |
| OSC_study48 | 0.55 | 30 | 0.01 | 0.49 | 0.95 | 1 |
| OSC_study49 | 0.55 | 33 | -0.01 | 0.52 | 0.96 | 1 |
| OSC_study50 | 0.57 | 41 | -0.05 | 0.63 | 0.99 | 1 |
| OSC_study51 | 0.62 | 13 | 0.14 | 0.27 | 0.77 | 0.98 |
| OSC_study52 | 0.62 | 280 | -0.23 | 0.98 | 1 | 1 |
| OSC_study53 | 0.63 | 33 | -0.05 | 0.62 | 0.99 | 1 |
| OSC_study54 | 0.63 | 39 | -0.07 | 0.68 | 0.99 | 1 |
| OSC_study55 | 0.65 | 32 | -0.05 | 0.63 | 0.99 | 1 |
| OSC_study56 | 0.68 | 15 | 0.08 | 0.36 | 0.89 | 1 |
| OSC_study57 | 0.69 | 25 | -0.03 | 0.57 | 0.98 | 1 |
| OSC_study58 | 0.69 | 19 | 0.02 | 0.45 | 0.95 | 1 |
| OSC_study59 | 0.71 | 25 | -0.04 | 0.59 | 0.99 | 1 |
| OSC_study60 | 0.73 | 38 | -0.12 | 0.78 | 1 | 1 |
| OSC_study61 | 0.78 | 16 | 0.02 | 0.46 | 0.96 | 1 |
| OSC_study62 | 0.81 | 23 | -0.08 | 0.66 | 1 | 1 |
| OSC_study63 | 0.83 | 126 | -0.28 | 0.99 | 1 | 1 |
| OSC_study64 | 0.84 | 30 | -0.13 | 0.79 | 1 | 1 |
| OSC_study65 | 0.86 | 30 | -0.15 | 0.81 | 1 | 1 |
| OSC_study66 | 0.87 | 11 | 0.08 | 0.37 | 0.92 | 1 |
| OSC_study67 | 0.9 | 22 | -0.1 | 0.71 | 1 | 1 |
| OSC_study68 | 0.9 | 101 | -0.3 | 0.99 | 1 | 1 |
| OSC_study69 | 0.91 | 9 | 0.14 | 0.31 | 0.87 | 1 |
| OSC_study70 | 0.91 | 15 | -0.03 | 0.55 | 0.99 | 1 |
| OSC_study71 | 0.95 | 9 | 0.13 | 0.33 | 0.89 | 1 |
| OSC_study72 | 1.02 | 78 | -0.34 | 1 | 1 | 1 |
| OSC_study73 | 1.29 | 9 | -0.01 | 0.52 | 0.99 | 1 |

*Note*: The effect size of the original study with

* is Fisher z transformed partial correlation. All other effect sizes are Fisher z transformed correlations.

The seventh column in Table 3 represents the probability of finding at least the minimal replication effect size for the 17 original SCORE studies, given Model 2 (i.e., the moderate estimate model, where the true effect size $\sim N(0.5*$effect size original study, 0.1)). Here, we can observe that all probabilities are higher than 50%, and 13 out of 17 original SCORE studies have a probability of more than 90% to be successfully replicated under the moderate model. Looking at the blue circles in Fig 2 (and column 6 in Table 4), we can observe that almost all probabilities are higher than 50% (71 out of 73), and approximately half of the original OSC [4] studies (35 out of 73) have a probability of more than 90% to be successfully replicated, given the moderate model. Thus, if the true population effect sizes are half of the originally reported sample effect size, the conclusion of a successful replication is very likely when assessed through meta-analysis.

Finally, column 8 in Table 3 contains the probability of finding at least the minimal replication effect size for the 17 original SCORE studies, given the optimistic model (i.e., the true effect size $\sim N($effect size original study, 0.1)). It can be seen that all probabilities are almost equal to 100%, meaning that according to our analysis, all original SCORE studies have a

probability of almost 100% to be successfully replicated, given the original study-based estimate model. For the OSC [4] data, almost all probabilities (except for 5 probabilities) are almost equal to 100%, so almost all original OSC [4] studies have a probability of almost 100% to be successfully replicated, given the original study-based estimate model (see the purple circles in Fig 2 and column 7 in Table 4). Thus, if the true population effect sizes are equal to the originally reported sample effect size, the conclusion of a successful replication is almost certain when assessed through meta-analysis.

Accuracy of the predicted overall replication success was evaluated by comparing the true and predicted percentage of replication successes. For each model separately, we calculated the (adjusted) Brier scores, which can be found in Table 5 (the results for SCORE replication studies can be found in S1 Table). We performed this calculation for replication success quantified as (1) replication study is significant, and the replication effect size is in the same direction as the original effect size and (2) meta-analysis is significant, and the meta-analytic effect size is in the same direction as the original effect size. Based on the first replication success method, it can be concluded that only the sceptic model performed better than chance (11% between chance and the best possible score); the moderate and optimistic models performed worse than chance (20% and 28% between chance and the worst possible score respectively). This means when the population effect sizes were centered around zero, the prediction was more accurate compared to the prediction when population effect sizes were centered around (half) the original sample effect size. The prediction was slightly less bad when population effect sizes were centered around half the original sample effect size than when centered around the original sample effect size. Overall, it can be concluded that none of the three models predicted very well.

Brier scores for replication success quantified using meta-analysis were much better. Table 5 shows that the sceptic, moderate, and optimistic models scored 34%, 25%, and 13% between chance and perfect performance respectively. The relative ordering in performance of the models reflects that of replication success as quantified based on using the first replication success approach: the sceptic model appears to be most accurate.

For the OSC [4] studies, the Brier scores were calculated as well. In Table 5, the (adjusted) Brier scores can be found for each model separately (the results for OSC [4] replication studies can be found in S2 Table). When replication success was quantified based on the $p$-value and the direction of the effect size, it can be concluded that the scores were at chance performance for the sceptic model. The moderate and optimistic model both performed below chance (37% and 54% between chance and the worst possible score respectively).

**Table 5. Brier and adjusted Brier scores per model for SCORE and OSC [4] data.**

| Models | SCORE: $p$ and direction | | SCORE: MA | | OSC: $p$ and direction | | OSC: MA | |
|---|---|---|---|---|---|---|---|---|
| | Brier score | adjusted Brier score | Brier score | adjusted Brier score | Brier score | adjusted Brier score | Brier score | adjusted Brier score |
| Model 1: sceptic | 0.20 | 0.11 | 0.11 | 0.34 | 0.25 | 0.00 | 0.24 | 0.02 |
| Model 2: moderate | 0.36 | -0.20 | 0.14 | 0.25 | 0.47 | -0.37 | 0.18 | 0.15 |
| Model 3: optimistic | 0.41 | -0.28 | 0.19 | 0.13 | 0.59 | -0.54 | 0.22 | 0.06 |

*Note*: "$p$ and direction" refers to the (adjusted) Brier scores when replication success was quantified as a significant effect in the replication study combined with the replication effect size being in the same direction as the original effect size (i.e., positive). "MA" refers to the (adjusted) Brier scores when replication success was quantified as a significant meta-analytic result combined with the meta-analytic effect size being in the same direction as the original effect size (i.e., positive). The (adjusted) Brier scores for "SCORE: MA" are calculated based on 15 rather than 17 SCORE studies; two studies were excluded because the replication study had a different outcome measure than the original study. The Brier score can have a value between 0 and 1, and the smaller the Brier score, the more accurate the prediction. The adjusted Brier score can have a value between -1 and +1, and the larger the adjusted Brier score, the more accurate the prediction.

When replication success was quantified using meta-analysis, performance of all three models improved, but the relative order changed: The moderate model performed best (15% between chance and perfect), followed by the optimistic model (6% between chance and perfect), with the sceptic model performing worst (2% between chance and perfect). As such, when viewed through a meta-analytic lens, it seems the moderate model best reflects the underlying population effect sizes for the studies incorporated in OSC [4].

We have repeated the entire procedure described above for two additional scenarios in which we varied the sample size of the generated replication studies. We did this in order to study the effect of the relative difference between the replication sample size and the original sample size on the probability of finding at least the minimal replication effect size given the three models. In one scenario the replication sample size was as large as the sample size of the original study (i.e., 1:1), and in the other scenario the replication sample size was five times larger than the original sample size (i.e., 5:1). The results for the three models can be found in S3 Table for the SCORE studies, and in S4 Table for the OSC [4] studies in the Supporting information.

In the 1:1 scenario, 15 out of 17 SCORE studies would be counted as replication success even if certain replication sample effect size lower than zero were obtained from a meta-analytic perspective. In the 5:1 scenario, 7 out of 17 SCORE studies with such outcomes would be counted as replication success. The corresponding numbers for the OSC [4] data are 44 and 9 out of 73 studies, respectively. It appears that the larger the sample size of the replication study is compared to the sample size of the original study, the less likely it is for the minimal replication effect size to be negative. However, it is important to note that the positive minimal replication effect sizes are generally close to zero. This means that replication studies with effect sizes close to zero would still be counted as successful.

Based on Model 1 (i.e., the sceptic estimate), 15 out of 17 SCORE studies have replication success probabilities higher than 50%, and the other 2 studies have probabilities higher than 30% (but lower than 50%) in the 1:1 scenario. In the 5:1 scenario, 7 out of 17 such probabilities are higher than 50%; 9 probabilities are higher than 30% (but lower than 50%), and only one is lower than 30%. For the OSC [4] studies, 45 out of 73 probabilities are higher than 50%, 25 are higher than 30% (but lower than 50%), and only 3 are lower than 30% in the 1:1 scenario. In the 5:1 scenario, 9 out of 73 probabilities are higher than 50%; 49 probabilities are higher than 30% (but lower than 50%), and 15 are lower than 30%. Based on these results it appears that the larger the sample size of the replication study is compared to the sample size of the original study, the fewer studies have a higher probability than 50% to be successfully replicated, given Model 1.

Based on Model 2 (i.e., moderate estimate), all SCORE studies have probabilities higher than 50%, and 14 studies have probabilities higher than 90% in the 1:1 scenario. In the 5:1 scenario, all probabilities are higher than 50%, and 15 out of 17 probabilities are higher than 90%. For the OSC [4] studies, 71 out of 73 probabilities are higher than 50%; 37 studies have probabilities higher than 90%, and only 2 studies have probabilities lower than 30% in the 1:1 scenario. In the 5:1 scenario, 71 out of 73 probabilities are higher than 50%; 36 probabilities are higher than 90%; no studies have a probability lower than 30%.

Based on Model 3 (i.e., optimistic estimate), all SCORE studies have probabilities (almost) equal to 100% both in the 1:1 and 5:1 scenarios. For the OSC [4] studies, 67 out of 73 studies have a probability higher than 90%; 4 studies have a probability higher than 50% (but lower than 90%); 2 studies have a probability lower than 30% in the 1:1 scenario. In the 5:1 scenario, 70 out of 73 studies have a probability higher than 90%, 1 has a probability smaller than 50%, and none have a probability smaller than 30%.

Based on these results, it appears that the relative difference between the sample size of the replication studies and the sample size of the original studies does not seem to influence the probability of replication success much, given Model 2 or 3.

The (adjusted) Brier scores can be found in S5 Table for the SCORE studies, and in S6 Table for the OSC [4] studies in the Supporting information.

For the SCORE studies, when replication success was quantified based on the *p*-value and the direction of the effect size in the 1:1 scenario, it can be concluded that the scores were at chance performance for the sceptic model. The moderate and optimistic model both performed below chance (20% and 27% between chance and worst possible score respectively). In the 5:1 scenario, the sceptic model had the best performance (12% between chance and perfect); the moderate and optimistic models performed again below chance (22% and 28% between chance and worst possible score).

When replication success was quantified using meta-analysis, the performance of all three models improved to a large extent and the order remained the same in the 1:1 scenario: the sceptic model performed best (47% between chance and perfect), followed by the moderate and optimistic models (25% and 13% between chance and perfect respectively). In the 5:1 scenario, the relative order changed: the moderate model performed best (21% between chance and perfect), followed by the optimistic model (11% between chance and perfect), and the sceptic model performed the worst (9% between chance and perfect).

For the OSC [4] studies, when replication success was quantified based on the *p*-value and the direction of the effect size in the 1:1 scenario, all models performed below chance (sceptic 11%, moderate 39%, and optimistic 53% between chance and worst). In the 5:1 scenario, the sceptic model had the best performance (3% between chance and perfect); the moderate and optimistic models performed again below chance (39% and 55% between chance and worst possible score).

When replication success was quantified using meta-analysis, the performance of all three models improved, but the relative order changed: in the 1:1 scenario, the moderate model performed best (15% between chance and perfect), followed by the sceptic and optimistic models (13% and 7% between chance and perfect). In the 5:1 scenario, the relative order changed again: the moderate model performed best again (16% between chance and perfect), followed by the optimistic model (6% between chance and perfect), and the sceptic model performed below chance (12% between chance and worst).

## Discussion

Although meta-analysis has not been developed to evaluate replication success, in practice, it has been used in this context. Therefore, in the present study, we used 17 original SCORE studies and 73 original OSC [4] studies to get more insight into how meta-analysis functions as a replication success metric. We did this by making aggregate predictions regarding replicability under different assumptions about the population when including the original study and its single replication in the meta-analysis. We included the following assumptions regarding the population: a model where the underlying population effect is zero (sceptic), a model where the underlying population effect is half of the originally reported sample effect size (moderate), and a model where the underlying population effect is equal to the originally reported sample effect size (optimistic). Afterwards, accuracy of the predicted overall replication success was evaluated by comparing the true and predicted percentage of replication successes.

The take-home message based on the results found in the present study is that if the underlying population effect is zero, the replications still have a high likelihood of being successful when viewed through a meta-analytic lens, unless the sample size of the replication study is

much larger than the sample size of the original study (the highest ratio investigated in the present study was 5:1); this conclusion is even more likely if the true population effect size is half of the originally reported sample effect size; if the true population effect size is equal to the originally reported sample effect size, the conclusion of a successful replication is almost certain when assessed through meta-analysis.

The accuracy of the predictions was dependent on how replication success was quantified in the observed replication studies. When replication success in the observed replication studies was quantified based on the *p*-value and the direction of the effect size, the predictions were fairly poor. When replication success in the observed replication studies was quantified based on meta-analysis, the predictions were better. This does make sense because in the latter, the same method (i.e., meta-analysis) was used in predicting replication success and in quantifying replication success for the observed replication studies. The sceptic model appeared to be the most accurate for the SCORE studies, except in the 5:1 scenario when replication success was based on meta-analysis: in this case the moderate model appeared to be the most accurate. For the OSC [4] studies, when replication success was quantified based on the *p*-value and the direction of the effect size, the scores were around chance performance for the sceptic model in the 2:1 and 5:1 scenarios, but below chance performance in the 1:1 scenario. The moderate and optimistic model both performed below chance. When replication success was quantified using meta-analysis, the moderate model performed best in all three replication sample size scenarios. Thus, the models that predict true effect sizes between zero and half the effect size of the original study map best onto the empirical replication results.

A limitation of the present study is that when we evaluated the accuracy of the predicted overall replication success, we compared the predicted percentage of replication success with the true percentage of replication success. However, no objective measure for quantifying replication success exists. In the present study, the true percentage of replication success was based on 1) the evaluation of a significant result in the replication study, and the replication effect size being in the same direction as the original effect size, and 2) on the evaluation of a significant meta-analytic result, and the meta-analytic effect size being in the same direction as the original effect size. However, the outcomes based on these approaches do not represent the true percentage of replication success because there is no replication success metric that functions perfectly [e.g., 13].

Another limitation is that the included number of studies between the two projects was vastly different (SCORE project: n = 17; OSC project: n = 73). Although we do not think that this issue necessarily affected the results in the present study, because the conclusions based on both projects were similar, we cannot rule this out. A related issue is that the 17 original SCORE studies were from diverse disciplines within the social sciences, while the OSC originals were all psychology studies. This heterogeneity could potentially have affected our results.

If one is interested in using meta-analysis in the context of replication research in which one wants to evaluate whether the replication result is in line with the original result, then we think that meta-analysis is an inappropriate tool. This is because meta-analytic results represent a combination of the results of the replication study and the original study. This comparison might seem meaningless to some extent because replication researchers are often interested in studying whether an independent replication study leads to the same conclusion as the original study. In case of meta-analysis, there is dependency, as one compares the results of the original study to results that are a combination of results obtained from the replication study and from the original study.

Furthermore, it is important to take publication bias into account in the context of meta-analysis. Combining original published findings with new replication findings by means of meta-analytic techniques ignores publication bias. In a recent review by Bartoš et al. [34], the

extent of the issue of publication bias in various scientific fields was found to be substantial. As such, synthesizing the findings propagates the bias in the original finding. The resulting meta-analytic summary becomes necessarily biased, and it becomes difficult to find a replication unsuccessful. The fact that findings chosen for replication are almost always significant (e.g., 97% of the original findings in the OSC [4] project had a statistically significant *p*-value) makes this issue even more severe. An interesting method that takes publication bias into account is the Bayesian averaging model proposed by Guan and Vandekerckhove [35]. This approach leads to more conservative interpretations of meta-analyses.

It is important to note that it does not mean that meta-analysis is meaningless in general. If one is interested in combining the results of multiple studies for an overall assessment regarding a certain phenomenon (which is the purpose of meta-analysis), then we think that meta-analysis is an appropriate approach. However, it is important to remain cautious about more general problems that might occur when using meta-analysis for this purpose. These issues have been discussed by, for example, Senn [36] and Simonsohn, Simmons, and Nelson [37].

Conducting replication studies in science has been perceived as highly important [e.g., 2, p. 2]. However, the quantification of replication success is complicated by the fact that there is no golden standard in quantifying replication success [e.g., 4]. Additionally, replication success has often been used in a binary form. We believe this binary form is too simplistic, and we suggest that focusing on the degree of similarity between two studies is more appropriate [e.g., 13]. Furthermore, it is important to note that hidden moderators (i.e., contextual differences across the original and the replication study [e.g., 38]) could play a role in the differences in the findings between the original and replication study.

Based on the results obtained in the present study, we can conclude that meta-analysis mostly seems to be a liberal method when using it as a replication success metric. Meta-analysis often results in a high probability that the original study will be successfully replicated, especially in a world in which most published results happen to indicate a non-zero population effect, and in which publication bias is arguably still relatively common.

## Supporting information

**S1 Table. Results for SCORE replication studies.**
(XLSX)

**S2 Table. Results for OSC [4] replication studies.**
(XLSX)

**S3 Table. Results for the three models for the SCORE studies.**
(XLSX)

**S4 Table. Results for the three models for the OSC [4] studies.**
(XLSX)

**S5 Table. (Adjusted) Brier scores for the SCORE studies.**
(XLSX)

**S6 Table. (Adjusted) Brier scores for the OSC [4] studies.**
(XLSX)

## Author Contributions

**Conceptualization:** Jasmine Muradchanian, Rink Hoekstra, Henk Kiers, Don van Ravenzwaaij.

**Formal analysis:** Jasmine Muradchanian, Rink Hoekstra, Henk Kiers, Don van Ravenzwaaij.

**Investigation:** Jasmine Muradchanian, Rink Hoekstra, Henk Kiers, Don van Ravenzwaaij.

**Methodology:** Jasmine Muradchanian, Rink Hoekstra, Henk Kiers, Don van Ravenzwaaij.

**Project administration:** Jasmine Muradchanian.

**Supervision:** Rink Hoekstra, Henk Kiers, Don van Ravenzwaaij.

**Writing – original draft:** Jasmine Muradchanian.

**Writing – review & editing:** Jasmine Muradchanian, Rink Hoekstra, Henk Kiers, Don van Ravenzwaaij.

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
