## [Decision Letter · Decision Letter 0]

7 Jun 2024

PONE-D-24-14525Evaluating meta-analysis as a replication success measurePLOS ONE

Dear Dr. Muradchanian,

Thank you for submitting your manuscript to PLOS ONE. After careful consideration, we feel that it has merit but does not fully meet PLOS ONE’s publication criteria as it currently stands. Therefore, we invite you to submit a revised version of the manuscript that addresses the points raised during the review process.

We look forward to receiving your revised manuscript.

Kind regards,

Danka Purić, PhD

Academic Editor

PLOS ONE

Additional Editor Comments:

Dear authors,

Like the reviewers, I have also found your manuscript a very interesting read and I think it will be valuable and informative to anyone interested in issues of replicability and meta-science in general. Apart from the comments provided by the reviewers, I have only a few small points to add.

On lines 259-261 you say "Replication success was established if the meta-analytic result was significant (i.e., p < .05), and the meta-analytic effect size was in the same direction as the original effect size." Since effect sizes used in your MA have previously been transformed to their absolute values, this may sound a little confusing. Perhaps you could clarify that these were all positive.

In Tables 3 and 4, could you add a study identifier (even just study01, study 02...), to make it more evident what the rows of the table are. Since you are using studies from two projects, perhaps you could preface these identifiers with project names (e.g. SCORE_study01, OSC_study01).

Finally, even though the SCORE data are unavailable at the moment, it would be helpful to direct the readers to the SCORE website in your data availability statement. Through the website the readers could contact the coordinators for access to the data, at least until it becomes publicly available. Even at this point, the website will still be a useful landing page for those interested in accessing the datasets.

Reviewers' comments:

Reviewer's Responses to Questions

**Comments to the Author**

1. Is the manuscript technically sound, and do the data support the conclusions?

Reviewer #1: Yes

Reviewer #2: Yes

2. Has the statistical analysis been performed appropriately and rigorously? 

Reviewer #1: Yes

Reviewer #2: Yes

3. Have the authors made all data underlying the findings in their manuscript fully available?

Reviewer #1: Yes

Reviewer #2: Yes

4. Is the manuscript presented in an intelligible fashion and written in standard English?

Reviewer #1: Yes

Reviewer #2: Yes

5. Review Comments to the Author

Reviewer #1: I enjoyed reading the manuscript. It addresses an important topic—quantification of replication success—and was well written. I have only a couple of comments and suggestions that I believe might focus and strengthen the message.

Major suggestions 

- Abstract: Based on the results of the investigation, I think that authors should be more forceful in recommending against using meta-analyses as a metric of replication success. All the presented evidence shows that meta-analyses are unfit to be used as such.

- I would recommend dropping the accuracy of replications evaluation from the manuscript. Unless I'm missing something, it uses the same method as a predictor and a criterion. This is well disclosed in the manuscript and discussion, but it does not add much value to the manuscript. (I feel a bit sorry for the authors as collecting and analyzing the data must have been considerable work, but the manuscript could be more persuasive and impactful without the corresponding section. Simply showing how large and different replication study findings must be to warrant replication unsuccessful would make the manuscript more focused.) Consequently, multiple sections could be dropped from the manuscript making it more focused,  e.g., lines 96-125, 252-284, 353-395, ...). Some of the OSC/Replicats data could be used to show how hilariously impossible it is to obtain a non-significant meta-analytic effect if the null hypothesis was true.)

Minor comments:

- Synthesizing original published findings and a new RRR using a standard meta-analytic technique completely ignores the (probably substantial; see, e.g., Bartos et al., 2024 for a recent review) publication bias. As such, synthesizing the findings propagates the bias in the original finding. The resulting meta-analytic summary becomes necessarily biased, and it becomes practically impossible to find a replication unsuccessful. I think this point could be stressed more. (Also, the fact that we are usually replicating statistically significant findings makes this issue worse.)

- There is a plethora of publication bias correction methods that try to address publication bias when combining multiple published estimates. However, they are unfit for the current task as they require multiple estimates and assume all findings are potentially affected by publication bias. Nevertheless, the meta-analytic model of Guan and Vandekerckhove (2016) was proposed for very similar settings (Simulation studies-1) is worth mentioning.

- Lines 90-95 are indeed some problems for meta-analyses but not relevant for the examined application of meta-analyses. I recommend removing them as the description is distracting. 

Bartoš, F., Maier, M., Wagenmakers, E. J., Nippold, F., Doucouliagos, H., Ioannidis, J. P., ... & Stanley, T. D. (2024). Footprint of publication selection bias on meta‐analyses in medicine, environmental sciences, psychology, and economics. Research Synthesis Methods.

Guan, M., & Vandekerckhove, J. (2016). A Bayesian approach to mitigation of publication bias. Psychonomic Bulletin & Review, 23(1), 74-86.

Reviewer #2: The authors report a study where they examine the utility of using meta-analysis as a measure of replication success (or failure). Using a high-known study looking at the replicability of psychology studies and a new, yet to be reported study, SCORE, they examine how different model assumptions hold up to using meta-analytic combination of an original and replication study to determine if a replication was successful (or not).

Overall, this is a well-written study on a topic that is of importance across disciplines. The authors describe their study, with limitations, well. Below are suggestions on how to improve the readability of the study for the authors to consider.

Major comment:

My main comment is on the selection of SCORE and RPP for this study? The rationale was not stated (unless I missed it) and I think considering one (RPP) is published and one is not (SCORE) is a key point that the authors do not discuss. Related, is that the included samples of each study are vastly different (RPP – 100 studies with 73 included, SCORE – 81 with 17 included) – though this aspect is addressed by effect size type.

My recommendation is for the authors to consider presenting that these two studies they are leveraging are different and how the authors are leveraging them. For example, it seems, though is not stated, that RPP is a larger sample sized study the authors could leverage to test their research questions though is flawed because outcomes are known. This would be true of other studies referenced in the introduction. This seems like a good use of an existing study to examine their assumptions, knowing they have a bias they need to control – mainly that they know the outcomes of replications (and meta-analyses of original and replications) are known. However, there is a good sample size to lean on. SCORE on the other hand has a smaller usable sample size (see minor comment below about this) but crucially replication (and meta-analysis) outcomes are unknown to the authors at the time of developing their models. This has the advantage of a ‘truer’ test of the authors questions and suffers from sample size limitations.

I wonder if the authors would be open to presenting this more like how many psychology studies would present two studies – one on RPP and one about SCORE? I’m not sure if this would misrepresent how they conducted their investigation, so defer to the authors on this, but from a reader perspective it is hard to tell why these two studies were chosen, the pros/cons of each, and considering what I wrote above, the strength of inference (and by extension the limitations of inference) each study they leverage for this investigation has. So even if not presented as two studies, since that might be misrepresenting what occurred, presenting as two use cases might be useful. If so, I’d recommend anchoring first on RPP since it’s well known and has a large sample size, and then SCORE becomes the strong test case where outcomes are not publicly known and the authors own knowledge of outcomes is unknown.

Minor comments:

What are the disciplines of the 17 SCORE studies used? Are they psychology or other disciplines? This would be worth noting in the methods and potentially in the limitations section.

Do the sample sizes between RPP and SCORE limit what conclusions the authors can draw from their studies? Maybe this is a limitation of this study to add in the discussion?

I wonder if for Fig 2 the authors can add fitted curves to the plots? This would help draw the reader’s attention to how to interpret the information. It’s similar to graphs that show power as a function of effect size and sample size.

Do results change if a random-effects model is used instead of fixed? Related, did you try variations in sample size ratios (e.g., in addition to a 2:1 replication:original sample size, what does it look like for a 5:1 (such as in Camerer et al., 2016) or a 1:1?

Line 239 – it should be ‘the’ instead of ‘het’

6. PLOS authors have the option to publish the peer review history of their article (what does this mean?). If published, this will include your full peer review and any attached files.

Reviewer #1: No

Reviewer #2: **Yes: **Timothy M. Errington

---

## [Author Response · Author response to Decision Letter 0]

20 Jul 2024

Editor 

Like the reviewers, I have also found your manuscript a very interesting read and I think it will be valuable and informative to anyone interested in issues of replicability and meta-science in general. Apart from the comments provided by the reviewers, I have only a few small points to add.

E-0: We would like to thank you and the reviewers for reading our manuscript and for providing us with feedback. 

On lines 259-261 you say "Replication success was established if the meta-analytic result was significant (i.e., p < .05), and the meta-analytic effect size was in the same direction as the original effect size." Since effect sizes used in your MA have previously been transformed to their absolute values, this may sound a little confusing. Perhaps you could clarify that these were all positive.

E-1: We agree that the section regarding the meta-analytic effect size being in the same direction as the original effect size might confuse the reader, since we have transformed the original effect sizes to their absolute values. Therefore, we have rewritten this section as (see underlined part):

“Replication success was established if the meta-analytic result was significant (i.e., p < .05), and before transformation, the meta-analytic effect size was in the same direction as the original effect size.” 

In Tables 3 and 4, could you add a study identifier (even just study01, study 02...), to make it more evident what the rows of the table are. Since you are using studies from two projects, perhaps you could preface these identifiers with project names (e.g. SCORE_study01, OSC_study01).

E-2: We have now added study identifiers with a preface referring to the project names in Tables 3 and 4 (i.e., SCORE_study.. for the SCORE studies, and OSC_study.. for the OSC studies).

Additionally, we have added the following sentence to the first paragraph in the Results section:

“The first column in Table 3 represents a study identifier.”

Finally, we have adjusted the column number references to Tables 3 and 4 in the text in the Results section.

Finally, even though the SCORE data are unavailable at the moment, it would be helpful to direct the readers to the SCORE website in your data availability statement. Through the website the readers could contact the coordinators for access to the data, at least until it becomes publicly available. Even at this point, the website will still be a useful landing page for those interested in accessing the datasets.

E-3: We have now added the following information to the “Data and code availability” section:

“For accessing the SCORE data, the reader can contact the coordinators of the SCORE program through the following website: https://www.cos.io/score.”

Reviewer #1

I enjoyed reading the manuscript. It addresses an important topic—quantification of replication success—and was well written. I have only a couple of comments and suggestions that I believe might focus and strengthen the message.

R1-0: Thank you very much for reading our manuscript and for your comments. 

Major suggestions 

- Abstract: Based on the results of the investigation, I think that authors should be more forceful in recommending against using meta-analyses as a metric of replication success. All the presented evidence shows that meta-analyses are unfit to be used as such.

R1-1: We agree that meta-analysis is an inappropriate tool to measure replication success, so we have rephrased the last sentence of the abstract (see the underlined part below):

“Furthermore, we argue that there are fundamental reasons against using meta-analysis as a metric for replication success.”

- I would recommend dropping the accuracy of replications evaluation from the manuscript. Unless I'm missing something, it uses the same method as a predictor and a criterion. This is well disclosed in the manuscript and discussion, but it does not add much value to the manuscript. (I feel a bit sorry for the authors as collecting and analyzing the data must have been considerable work, but the manuscript could be more persuasive and impactful without the corresponding section. Simply showing how large and different replication study findings must be to warrant replication unsuccessful would make the manuscript more focused.) Consequently, multiple sections could be dropped from the manuscript making it more focused, e.g., lines 96-125, 252-284, 353-395, ...). Some of the OSC/Replicats data could be used to show how hilariously impossible it is to obtain a non-significant meta-analytic effect if the null hypothesis was true.)

R1-2: We agree with the reviewer that one of the main insights of our paper is to show, based on data, the relationship between sample size and the odds of getting a successful replication. We also, however, believe that there is merit to showing what happens when combining two studies. Meta-analysis is not a very useful measure of replication success, but we think it can be a useful method to show the overall effect of two closely comparable studies. Thus, we would prefer to leave the parts in that the reviewer suggested to leave out.

Minor comments:

- Synthesizing original published findings and a new RRR using a standard meta-analytic technique completely ignores the (probably substantial; see, e.g., Bartos et al., 2024 for a recent review) publication bias. As such, synthesizing the findings propagates the bias in the original finding. The resulting meta-analytic summary becomes necessarily biased, and it becomes practically impossible to find a replication unsuccessful. I think this point could be stressed more. (Also, the fact that we are usually replicating statistically significant findings makes this issue worse.)

R1-3: Thank you for mentioning this point. We completely agree, so we are discussing this issue now in the Discussion section in the following way: 

“Furthermore, it is important to take publication bias into account in the context of meta-analysis. Combining original published findings with new replication findings by means of meta-analytic techniques ignores publication bias. In a recent review by Bartoš et al. [37], the extent of the issue of publication bias in various scientific fields was found to be substantial. As such, synthesizing the findings propagates the bias in the original finding. The resulting meta-analytic summary becomes necessarily biased, and it becomes difficult to find a replication unsuccessful. The fact that findings chosen for replication are almost always significant (e.g., 97% of the original findings in the OSC [4] project had a statistically significant p-value) makes this issue even more severe.”

- There is a plethora of publication bias correction methods that try to address publication bias when combining multiple published estimates. However, they are unfit for the current task as they require multiple estimates and assume all findings are potentially affected by publication bias. Nevertheless, the meta-analytic model of Guan and Vandekerckhove (2016) was proposed for very similar settings (Simulation studies-1) is worth mentioning.

R1-4: We agree that the study of Guan and Vandekerckhove (2016) is worth mentioning in our manuscript. We have added the following part to the Discussion section:

“An interesting method that takes publication bias into account is the Bayesian averaging model proposed by Guan and Vandekerckhove [38]. This approach leads to more conservative interpretations of meta-analyses.”

- Lines 90-95 are indeed some problems for meta-analyses but not relevant for the examined application of meta-analyses. I recommend removing them as the description is distracting.

R1-5: Agreed. We have removed this part from the manuscript.

Reviewer #2

The authors report a study where they examine the utility of using meta-analysis as a measure of replication success (or failure). Using a high-known study looking at the replicability of psychology studies and a new, yet to be reported study, SCORE, they examine how different model assumptions hold up to using meta-analytic combination of an original and replication study to determine if a replication was successful (or not).

Overall, this is a well-written study on a topic that is of importance across disciplines. The authors describe their study, with limitations, well. Below are suggestions on how to improve the readability of the study for the authors to consider.

R2-0: Thank you very much for reading our manuscript and for your comments.

Major comment:

My main comment is on the selection of SCORE and RPP for this study? The rationale was not stated (unless I missed it) and I think considering one (RPP) is published and one is not (SCORE) is a key point that the authors do not discuss. Related, is that the included samples of each study are vastly different (RPP – 100 studies with 73 included, SCORE – 81 with 17 included) – though this aspect is addressed by effect size type.

R2-1: Thank you for this suggestion. We agree that the rationale behind the idea of choosing SCORE and RPP was not clearly stated in our manuscript. We were interested in studying how meta-analysis functions as a replication success measure based on studies that have truly been replicated, without having access to the replication results in order to avoid our personal biases. Therefore, we decided to use the original studies in the SCORE project. Because we were ultimately only able to use 17 original SCORE studies, we thought that this number was somewhat small and decided to use the same approach on the RPP studies. We have now added the following (underlined) information to the Introduction section, and we hope that the rationale is more clearly stated:

“For each original SCORE study, we calculated the probability that the replication study would lead to a successful replication under different assumptions about the population. Note that replication results were not yet available so as not to influence the models we proposed for prediction. The probability of replication success can be perceived as a kind of statistical power (a meta-analysis replication power), conditional on the results of the original study. A successful replication in the present context means that the meta-analytic result is statistically significant (i.e., p < .05), and that the meta-analytic effect size is in the same direction as the original effect size. Accuracy of the predicted overall replication success was evaluated by comparing the observed and predicted percentage of replication successes. Because the number of studies we could use from the SCORE project was small, we decided to examine how well the models that we generated to predict replication success in the SCORE program would have predicted replication success for study results that were part of the OSC [4] project.”

My recommendation is for the authors to consider presenting that these two studies they are leveraging are different and how the authors are leveraging them. For example, it seems, though is not stated, that RPP is a larger sample sized study the authors could leverage to test their research questions though is flawed because outcomes are known. This would be true of other studies referenced in the introduction. This seems like a good use of an existing study to examine their assumptions, knowing they have a bias they need to control – mainly that they know the outcomes of replications (and meta-analyses of original and replications) are known. However, there is a good sample size to lean on. SCORE on the other hand has a smaller usable sample size (see minor comment below about this) but crucially replication (and meta-analysis) outcomes are unknown to the authors at the time of developing their models. This has the advantage of a ‘truer’ test of the authors questions and suffers from sample size limitations.

R2-2: We had decided to conduct the present study based on a sample of original studies that are part of the SCORE project because the replication results were not available to us in order to prevent our prediction models from being informed by the empirical outcomes. We have further clarified the reasons for using each of the data sets in the manuscript, as we described also in R2-1. 

I wonder if the authors would be open to presenting this more like how many psychology studies would present two studies – one on RPP and one about SCORE? I’m not sure if this would misrepresent how they conducted their investigation, so defer to the authors on this, but from a reader perspective it is hard to tell why these two studies were chosen, the pros/cons of each, and considering what I wrote above, the strength of inference (and by extension the limitations of inference) each study they leverage for this investigation has. So even if not presented as two studies, since that might be misrepresenting what occurred, presenting as two use cases might be useful. If so, I’d recommend anchoring first on RPP since it’s well known and has a large sample size, and then SCORE becomes the strong test case where outcomes are not publicly known and the authors own knowledge of outcomes is unknown.

R2-3: We understand the logic, but we actually operated in the reverse order. We first developed our prediction models, that we tested on a dataset for which empirical replication success was (at the time) unknown. Once we got to testing our models, we learned that the sample size we had available to us was smaller than we originally anticipated, so we decided to include an additional large-scale replication project. Because the prediction models had at that point already been established, there was no risk of the OSC data accidentally biasing our proposed models.

Minor comments:

What are the disciplines of the 17 SCORE studies used? Are they psychology or other disciplines? This would be worth noting in the methods and potentially in the limitations section.

R2-4: Thank you for mentioning this point. In the Method section, we have now added the disciplines of the 17 SCORE studies in the following way:

“The disciplines of these 17 studies were: Business & Economics; Environmental Sciences & Ecology; Psychology; Behavioral Sciences; Biomedical Social Sciences; Sociology.” 

We agree that the 17 SCORE studies were not only psychology studies (whereas the OSC studies were only psychology studies), and that this forms a potential limitation for our paper. Therefore, we have added the following part to the Discussion section:

“A related issue is that the 17 original SCORE studies were from diverse disciplines within the social sciences, while the OSC originals were all psychology studies. This heterogeneity could potentially have affected our results.”

Do the sample sizes between RPP and SCORE limit what conclusions the authors can draw from their studies? Maybe this is a limitation of this study to add in the discussion?

R2-5: We agree that the sample sizes between RPP and SCORE could limit our conclusions. Therefore, we have added this limitation to our Discussion section:

“Another limitation is that the included number of studies between the two projects was vastly different (SCORE project: n=17; OSC project: n=73). Although we do not think that this issue necessarily affected the results in the present study, because the conclusions based on both projects were similar, we cannot rule this out.”

I wonder if for Fig 2 the authors can add fitted curves to the plots? This would help draw the reader’s attention to how to interpret the information. It’s similar to graphs that show power as a function of effect size and sample size.

R2-6: Thank you for this point. We have now added fitted curves, based on logistic regressions, to the plots in Fig. 2.

Do results change if a random-effects model is used instead of fixed? Related, did you try variations in sample size ratios (e.g., in addition to a 2:1 replication:original sample size, what does it look like for a 5:1 (such as in Camerer et al., 2016) or a 1:1?

R2-7: We decided to use fixed effect models rather than random-effects models because the latter might cause overfitting if the number of studies included in the meta-analysis is smaller than 10 (see e.g., Scheibehenne et al., 2017).

We have ad

---

## [Editor Report · Decision Letter 1]

25 Jul 2024

Evaluating meta-analysis as a replication success measure

PONE-D-24-14525R1

Dear Dr. Muradchanian,

We’re pleased to inform you that your manuscript has been judged scientifically suitable for publication and will be formally accepted for publication once it meets all outstanding technical requirements.

Kind regards,

Danka Purić, PhD

Academic Editor

PLOS ONE

Additional Editor Comments (optional):

I would like to thank you for thoroughly addressing all of the reviewers' comments, as well as my own. The manuscript was very well written even before the first round of reviews, and it has now been significantly improved.

I agree with your decision to leave the information regarding accuracy in the manuscript, as I believe it may be beneficial for some readers. I also think the decision to first present the SCORE data and then the OSC data was warranted, as this is more true to the way the study was actually conducted. Importantly, this information is now also explicitly stated in the manuscript making your logic easier to follow.

As for the other comments - you have meticulously incorporated all suggestions and performed additional (quite informative) analyses, thus enriching your manuscript even further. From my perspective, a new round of reviews would be unlikely to provide you with important and constructive additional feedback - making your manuscript ready for publication as is.

All the best in your future work!
---

## [Editor Report · Acceptance letter]

5 Aug 2024

PONE-D-24-14525R1 

PLOS ONE

Dear Dr. Muradchanian, 

I'm pleased to inform you that your manuscript has been deemed suitable for publication in PLOS ONE. Congratulations! Your manuscript is now being handed over to our production team.

Kind regards, 

on behalf of

Dr. Danka Purić 

Academic Editor

PLOS ONE